

# I understand you feel that way, but I feel this way: the benefits of I-language and communicating perspective during conflict

Shane L. Rogers[1], Jill Howieson[2] and Casey Neame[1]

[1] Psychology Department, Edith Cowan University, Perth, WA, Australia
[2] Law School, University of Western Australia, Perth, WA, Australia

## ABSTRACT

Using hypothetical scenarios, we provided participants with potential opening statements to a conflict discussion that varied on I/you language and communicated perspective. Participants rated the likelihood that the recipient of the statement would react in a defensive manner. Using I-language and communicating perspective were both found to reduce perceptions of hostility. Statements that communicated both self- and other-perspective using I-language (e.g. *'I understand why you might feel that way, but I feel this way, so I think the situation is unfair'*) were rated as the best strategy to open a conflict discussion. Simple acts of initial language use can reduce the chances that conflict discussion will descend into a downward spiral of hostility.

## INTRODUCTION

During interpersonal conflict, the initial communication style can set the scene for the remainder of the discussion (*Drake & Donohue, 1996*). A psychological principle termed the *norm of reciprocity* describes a basic human tendency to match the behaviour and communication style of one's partner during social interaction (*Park & Antonioni, 2007*). During conflict, a hostile approach typically produces hostility in return from the other person, potentially creating a negative downward spiral (*Bowen, Winczewski & Collins, 2016*; *Park & Antonioni, 2007*; *Pike & Sillars, 1985*; *Wiebe & Zhang, 2017*). Therefore, it is not surprising that recommendations abound in the academic and popular literature about specific communication tactics to minimise perceptions of hostility (*Bloomquist, 2012*; *Hargie, 2011*; *Heydenberk & Heydenberk, 2007*; *Howieson & Priddis, 2015*; *Kidder, 2017*; *Moore, 2014*; *Whitcomb & Whitcomb, 2013*).

The present research assesses two specific aspects of language style that theorists have recommended as beneficial tactics for minimising hostility during conflict: the use of I-language instead of you-language, and communicating perspective (*Hargie, 2011*). We broadly define *communicating perspective* as language that is clearly communicating one's own point of view, and/or communicating an understanding of the perspective

Corresponding author
Shane L. Rogers,
shane.rogers@ecu.edu.au

of the other person. This paper sets out to examine the relative merits of I-language and communicating perspective, both singularly, and used together, to open communication to prevent conflict escalation.

## Communicating perspective—giving and taking

Consideration of the perspective of the other party is widely held to be beneficial during conflict (*Ames, 2008*; *Galinksy et al., 2008*; *Hargie, 2011*; *Howieson & Priddis, 2015*; *Kidder, 2017*). An understanding of perspectives facilitates a more integrative approach where parties are willing to compromise to arrive at a mutually beneficial solution (*Galinksy et al., 2008*; *Kemp & Smith, 1994*; *Todd & Galinsky, 2014*). Therefore, in dispute resolution a mediator will endeavour to encourage perspective taking by both parties (*Howieson & Priddis, 2012*, *2015*; *Ingerson, DeTienne & Liljenquist, 2015*; *Kidder, 2017*; *Lee et al., 2015*). However, fostering perspective taking is more involved than simply telling someone to try and consider the other person's point of view. In the negotiation/mediation literature it is recognised that increasing levels of perspective taking is best achieved *via the communication process* that occurs during a negotiation (*Howieson & Priddis, 2012*, *2015*; *Kidder, 2017*).

It is important to consider that perspective taking is an internal cognitive process, so is not readily apparent to the other party unless it is made observable via communication (*Kellas, Willer & Trees, 2013*). *Kellas, Willer & Trees (2013)* refer to this as *communicating perspective taking*, and report that married couples typically perceive agreement, relevant contributions, coordination, and positive tone as communicated evidence of perspective taking. A more *direct* strategy to communicate perspective taking is to paraphrase the perspective of the other party (*Howieson & Priddis, 2015*; *Kidder, 2017*). For example, by explicitly stating what you perceive is the other's point of view by saying something like 'What I'm hearing is that perhaps you aren't seeing enough evidence of appreciation and so feel like you're being taken for granted'. *Seehausen et al. (2012)* demonstrated the utility of paraphrasing by interviewing participants (interviewees) about a recent social conflict while varying interviewer responses across participants as simple note taking or paraphrasing. Interviewees receiving paraphrase responses reported feeling less negative emotion associated with the conflict compared to interviewees receiving note taking responses. Similarly, other research has reported that perceptions of *empathic effort* during conflict is associated with relationship satisfaction (*Cohen et al., 2012*).

While communicating perspective taking is useful during conflict, so is perspective giving (i.e. attempting to communicate one's own position/perspective) (*Bruneau & Saxe, 2012*; *Graumann, 1989*). For example, 'I am not feeling great about this because I don't feel like I am receiving a fair deal'. Perspective giving can be beneficial for the person offering the perspective to help them 'feel heard' (*Bruneau & Saxe, 2012*), and also assist the other party to engage in perspective taking, which fosters a greater sense of mutual understanding (*Ames, 2008*; *Ames & Wazlawek, 2014*). This is why negotiation experts recommend that during conflict both parties should communicate what their perspective is,

and also communicate explicitly to the other person that they are attempting to consider their perspective (*Howieson & Priddis, 2015*). However, while communication of perspective taking and giving have received research attention individually, no prior research has attempted to systematically compare them. In the present study we examine the perception of statements that vary in the extent of communicated perspective (i.e. none, self-only, other-only, self & other). By contrasting two types of communicating perspective (i.e. giving and taking) we can contribute to the research literature while also answering questions that have practical relevance for everyday communication.

## I-LANGUAGE AND YOU-LANGUAGE

Another aspect of language that is beneficial during conflict is the use of I-language (e.g. '*I think things need to change*') versus you-language (e.g. '*You need to change*') (*Hargie, 2011*; *Kubany et al., 1992a*; *Simmons, Gordon & Chambless, 2005*). For example, *Simmons, Gordon & Chambless (2005)* reported that a higher proportion of I-language and a lower proportion of you-language was associated with better problem solving and higher marital satisfaction. Similarly, *Bieson, Schooler & Smith (2016)* found that more frequent you-language during face-to-face conflict discussion was negatively associated with interaction quality of couples.

Kubany and colleagues took an experimental approach to directly assess the impact of I/you-language by examining participant ratings for statements that varied inclusion of I-language or you-language. Their research focused on the communication of emotion using I-statements (e.g. '*I am feeling upset*') versus you-statements (e.g. '*You have made me upset*'). In a series of studies it was found that I-language was less likely to evoke negative emotions and more likely to evoke compassion and cooperative behavioural inclinations in the recipient (*Kubany et al., 1992b*, *1995a*, *1995b*). The benefit of I-language over you-language is that I-language communicates to the recipient that the sender acknowledges they are communicating from their own point of view and therefore they are open to negotiation (*Burr, 1990*). Further, recipients often perceive you-language as accusatory and hostile (*Burr, 1990*; *Hargie, 2011*; *Kubany et al., 1992a*).

Furthermore, research from the embodied cognition field of study has produced evidence to suggest that text narratives using you-language foster more self-referential processing in the reader compared to I-language (*Beveridge & Pickering, 2013*; *Brunye et al., 2009*, *2011*; *Ditman et al., 2010*). Compared to I-language, studies have found you-language associated with subsequent faster response times for pictures from a self-perspective (versus other-perspective) (*Brunye et al., 2009*), better memory performance (*Ditman et al., 2010*), and higher emotional reactivity (*Brunye et al., 2011*). While these studies do not directly investigate conflict, the finding that you-language fosters greater self-referential processing is of relevance for conflict. As previously mentioned, during conflict the ideal scenario is for both parties to engage in mutual perspective taking to facilitate the search for a mutually beneficial solution. Therefore, you-language has potential to foster inward focus that can reduce the amount of perspective taking during communication.

### The present study

The present study will extend earlier research investigating the impact of I/you language and communicated perspective by employing a rating statements design similar to the experimental research of Kubany and colleagues (*Kubany et al., 1992a*, *1992b*, *1995a*, *1995b*). Based on prior research, we hypothesise that participants will rate: (1) I-language as less likely to provoke a defensive reaction than you-language (*Kubany et al., 1992a*, *1992b*; *Simmons, Gordon & Chambless, 2005*); and (2) communicating perspective as less likely to provoke a defensive reaction compared to statements that do not communicate any clear perspective (*Cohen et al., 2012*; *Gordon & Chen, 2016*; *Howieson & Priddis, 2015*; *Kellas, Willer & Trees, 2013*).

Importantly, we also varied the type of communicated perspective. As there has not been any prior research contrasting self-oriented versus other-oriented communicated perspective using a statement-rating paradigm, only tentative expectations were possible. We anticipated that a blend of one's own *and* the other's perspective would be received the best. We think this strategy is superior because it conveys both understanding (by acknowledging the other) and positive assertion (by acknowledging the self) (*Howieson & Priddis, 2015*).

## METHOD

### Participants

Participants were 253 university students (Mean = 28 years; SD = 8.75; 77% female). Prior to commencement of the research ethical approval was obtained from Edith Cowan University ethics committee (Ref: 15257). All participants supplied informed consent to take part in this research. Prior research by Kubany and colleagues using a similar rating statements paradigm reported significant effects with relatively low sample sizes ranging from 16 to 40 (*Kubany et al., 1992a*, *1992b*, *1995a*), with one study using a larger sample of 160 (*Kubany et al., 1995b*). We were therefore initially aiming to collect around 100 responses, but ended up with a larger sample, which has the benefit of assisting us feel more confident in our results.

### Materials

Six hypothetical scenarios were constructed that described a hypothetical conflict situation. One example scenario was: *Mike and his partner Lucy are living together and both working full time. Whenever Mike does some cleaning of the house and asks Lucy to help, Lucy typically replies that she is too tired after a full day at work to do cleaning. Mike feels it is unfair that he should be responsible for all the cleaning duties.*

Each scenario was designed with two protagonists, an offended party (e.g. Mike), and the party causing offense (e.g. Lucy). An overall problem was communicated within the scenario description (e.g. Lucy is not helping to clean the house as often as Mike would like). In addition, both the perspective of the offended party (e.g. Mike feels it unfair that he is responsible for all cleaning), and the offending party (e.g. Lucy is usually very tired after a full day at her work) were also included. After reading a scenario, the participant was presented with eight statements that the offended party might use to begin

**Table 1 Example statements provided to participants.**

| Statement type: | *Self & other perspective, with I-language.* |
|---|---|
| Example: | Lucy, I understand that you are very tired after work, but I feel it is unfair that I have to do all the cleaning by myself, and I think you should help with the cleaning. |
| Statement type: | *Self & other perspective, with you-language.* |
| Example: | Lucy, you are very tired after work, but it is unfair that I have to do all the cleaning by myself, and you should help with the cleaning. |
| Statement type: | *Other perspective, with I-language.* |
| Example: | Lucy, I understand that you are very tired after work, but I think you should help with the cleaning. |
| Statement type: | *Other perspective, with you-language.* |
| Example: | Lucy, you are very tired after work, but you should help with the cleaning. |
| Statement type: | *Self perspective, with I-language.* |
| Example: | Lucy, I feel that it is unfair that I have to do all the cleaning by myself, and I think you should help with the cleaning. |
| Statement type: | *Self perspective, with you-language.* |
| Example: | Lucy, it is unfair that I have to do all the cleaning by myself, and you should help with the cleaning. |
| Statement type: | *No perspective, with I-language.* |
| Example: | Lucy, I think you should help with the cleaning. |
| Statement type: | *No perspective, with you-language.* |
| Example: Y. | Lucy, you should help with the cleaning. |

Notes:
Provided above each statement in italics is the type of perspective/s communicated in the statement, and whether the statement is written predominately using either I or you language. Note that the statement type information provided in italics in the table was not presented to participants.

a conflict discussion with the offending party. These statements varied on the usage of I/you language and communicated perspective. Table 1 shows all the statements presented to participants for the example scenario.

The six scenarios were evenly spread across issues between romantic partners (two), friends (two), and work colleagues (two). All scenarios are provided in Supplemental Information 1 associated with this article. This research aims to investigate a general impact of subtle differences in communication rather than focusing on specific relationships or situations. The authors acknowledge that there is likely to be interesting differences to be found across different relationships and situations but those questions, while interesting, are beyond the scope of the present research.

## Procedure

Scenarios were presented in random order to participants, as were the order of the statements, via the online survey program Qualtrics. For each statement the participant rated the *likelihood that the offending party would react in a defensive manner* on a six-point scale (extremely unlikely to extremely likely).

## RESULTS

Participant ratings were averaged across scenarios. This provided a composite *likelihood of defensive reaction* score for each of the eight statement types, see Fig. 1. Next,

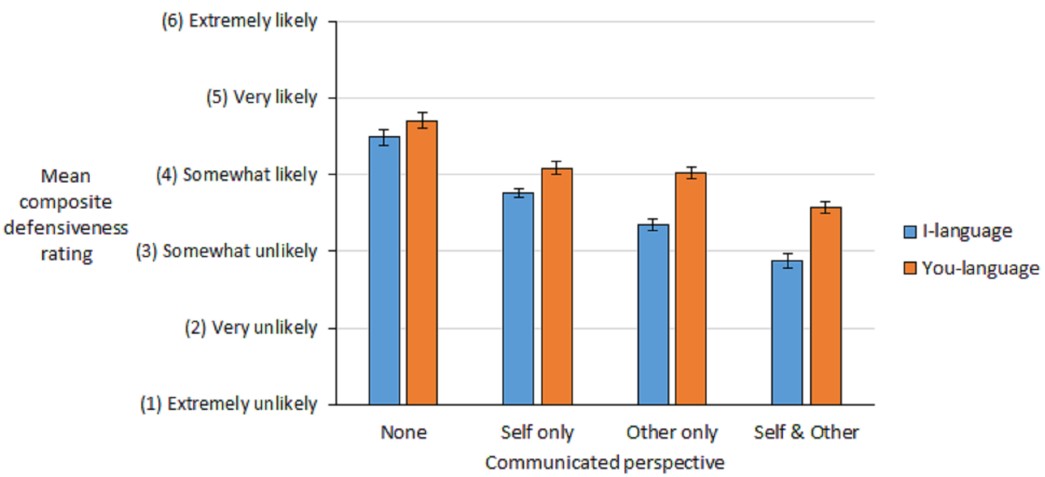

**Figure 1 Mean composite defensive rating scores for statements that varied in the type of communicated perspective, and I/you language; error bars represent 95% confidence limits.**

a 4 (communicated perspective: self and other, other-only, self-only, none) $\times$ 2 (I/you language: I-language, you-language) repeated measures factorial ANOVA was conducted on the mean composite scores. A main effect of I/you language was found, $F(1,252) = 357.88$, $p < 0.001$, $\eta_p^2 = 0.59$, demonstrated in Fig. 1 by the consistently lower defensiveness ratings for I-language statements compared to you-language statements. A main effect of communicated perspective was also found, $F(3,756) = 364.04$, $p < 0.001$, $\eta_p^2 = 0.59$. In Fig. 1 the difference in overall defensiveness ratings follows the pattern: none > self-only > other-only > self and other, all $p$s < 0.001.

A significant interaction effect was also found between I/you language and communicated perspective, $F(3,756) = 77.12$, $p < 0.001$, $\eta_p^2 = 0.23$. This interaction effect occurred because while there was a significant difference between I and you-statements for each perspective type (all $p$s < 0.001), the effect size was similar across self-and-other ($r = 0.72$), other-only ($r = 0.74$), and self-only ($r = 0.68$) perspective types but substantially lower when no perspective was communicated ($r = 0.47$). This suggests that when I-language is used in conjunction with communicated perspective there is a larger benefit of incorporating I-language compared to when no perspective is communicated.

## DISCUSSION

The present study investigated whether subtle changes in language can influence the perceived impact of statements used to open a conflict discussion. As expected, participants rated statements that contained I-language as having a lower likelihood of evoking a defensive reaction compared with statements that contained you-language. This result is consistent with earlier findings that report a superiority of I-language over you-language for conflict communication (*Bieson, Schooler & Smith, 2016*; *Kubany et al., 1992a, 1992b, 1995a, 1995b*; *Simmons, Gordon & Chambless, 2005*).

In the present study, the benefit of I-language compared to you-language was larger for statements that communicated one or more perspectives. For example, compared to simple statements such as *'Lucy, I think you should help with the cleaning'* versus *'Lucy, you should help with the cleaning'* there was a more pronounced benefit for I-language when the statements communicated an acknowledgement of the perspective of the offending party, such as *'Lucy, I understand that you are very tired after work, but I think you should help with the cleaning'* versus *'Lucy, you are very tired after work, but you should help with the cleaning'.*

Another expectation was that participants would rate communicating perspective more favourably compared to no communication of perspective. Results supported this expectation, which is consistent with prior recommendations that there is a benefit to communicating perspective during conflict and tense situations (*Ames, 2008*; *Cohen et al., 2012*; *Gordon & Chen, 2016*; *Howieson & Priddis, 2015*; *Kellas, Willer & Trees, 2013*; *Kidder, 2017*). Furthermore, the results revealed that when communicating perspective, the participants rated communicating both the self- *and* other-perspective as the most favourable, followed by communicating the other-perspective only, and finally the self-perspective only.

The present study is the first to directly compare ratings across statements that vary in the *type* of perspective communicated. These results suggest that as an individual act, communicating the perspective of the other is more important than communicating the perspective of the self. However, results also suggest that a combination of communicating *both* the perspective of the self and other is the best strategy for opening a conflict discussion.

In sum, the results suggest that we are more likely to receive a defensive and hostile reaction when we do not communicate any type of perspective, regardless of whether we use I-language or you-language. On the other hand, if we communicate using statements that include both the perspective of the self and the other person, and include I-language, then we are less likely to receive a defensive response. For example, the statement, *'Lucy, I understand that you are very tired after work, but I feel it is unfair that I have to do all the cleaning by myself, and I think you should help with the cleaning'*, which includes **both** self and other perspectives **and** I-language, was rated as the least likely to produce a defensive response.

## Limitations

Scholars have recognised that the non-interactive rating statements paradigm used in the present study limits the generalisability of findings beyond non-interactive communication contexts (*Bippus & Young, 2005*; *Kubany et al., 1992a*). Arguably, our findings reported here are more appropriately generalised to less interactive forms of communication such as text messaging and email. However, it must also be noted that the results of the present study do parallel findings investigating more interactive contexts (*Cohen et al., 2012*; *Gordon & Chen, 2016*; *Simmons, Gordon & Chambless, 2005*).

In our study we utilised a within-participants design, where each statement was rated by all participants. A within-participants approach for a rating statements task has been
criticized due to a potential risk that differences in the perception of statements might be exaggerated by participants engaging in a comparative process when making judgments (*Bippus & Young, 2005*). There is however no empirical evidence to confirm or disconfirm this assertion. In the present study, our interest was to gain insight into the relative utility of the statements used. So even if participants engaged in a comparative process we don't believe this seriously compromises the findings, as we were interested in determining which statement/s the participants thought was best (compared to all other statements). We concede however that there is a possibility that the magnitude of differences observed among statements in the present research might be lower if the study used a between-participants approach.

Our study is limited to perceptions regarding the likelihood of a defensive reaction. We assume that underlying the broader perception provided by our participants are more specific perceptions regarding levels of politeness, appropriateness, aggression, assertiveness, effectiveness, rationality, fairness, consideration, and clarity, and so on (*Bippus & Young, 2005*; *Hess et al., 1980*; *Kasper, 1990*; *Kubany et al., 1992b*; *Lewis & Gallois, 1984*; *Schroeder, Rakos & Moe, 1983*). The purpose of our study was to broadly assess the perception of the statements, so we decided upon broad terminology (i.e. rating the likelihood of a defensive response). Asking participants to rate all statements for many adjectives (e.g. aggression, clarity, and so on) would have caused the survey to be too long. Investigating the more specific perceptions mentioned here, perhaps using fewer scenarios and/or statements for practical reasons, is an avenue for future research.

## Future research

The present study suggests that I-language is less likely to produce a defensive reaction in a message recipient compared with you-language, particularly when it includes self and other communicated perspective. However, a related type of language framing outside the scope of the present study is *we-language.* Various researchers have suggested that we-language (e.g. '*We should talk about what we need to do to solve our problem*') is of greater benefit than I-language during times of conflict or tension (*Seider et al., 2009*; *Williams-Bausom et al., 2010*). *Seider et al. (2009)* argue that we-language emphasises togetherness and therefore fosters more collaboration compared to I-language, which might foster a sense of separation. A replication and extension of the present study incorporating statements predominately using we-language would therefore be a useful test of some arguments that exist for the benefits of we-language.

Our research has focused on *opening* statements to a conflict discussion. Assessing language for making *closing* statements would also be a useful avenue of further inquiry. Additionally, an investigation of the impact of specific statements *mid-discussion* would be interesting. For example, if the discussion has started badly, will the use of a targeted statement that communicates perspective using I-language 'fall on deaf ears'? Or, as discussed by *Howieson & Priddis (2015)*, might communicating perspective mid-discussion halt a downwards spiral, or even direct the flow of conversation into an upwards spiral? *Jameson, Sohan & Hodge (2014)* describe these kinds of moments in a negotiation as 'turning points'.

Another avenue for future research is to specifically investigate if communicating perspective, or I-language and you-language, have different effects across discursive actions such as requesting, describing, questioning, criticizing, blaming, offering alternatives, rejecting, refusing, inviting, and so on. As an example, consider the action of making an invitation (*Bella & Moser, 2018*; *Margutti & Galatolo, 2018*). What might be the best way to solicit a commitment? A simple request using you-language (e.g. *'You should come to dinner tomorrow night'*), or incorporating I-language (e.g. *'I think you should come to dinner tomorrow night'*), or communicating perspective in addition to the request (e.g. *'I think you should come to dinner tomorrow night, because I think you'll benefit from getting out of the house'* OR *'You should come to dinner tomorrow night, because you'll benefit from getting out of the house'*). The current research has potential to be extended to further understand how to best communicate during conflict, but also how to be more persuasive in general.

## CONCLUSION

A practical implication of this research is to provide empirical evidence to inform guidelines about how to frame opening statements during conflict. Generally, communicating some perspective (i.e. self and/or other) is better than neglecting to do so. When communicating perspective, the results of this study suggest that it is most beneficial to communicate *both* points of view (i.e. self and other) rather than a single perspective, using I-language.

This study has highlighted how different forms of language can interact with one another during conflict. Specifically, the study demonstrates how I-language becomes more beneficial for minimising hostility when one also communicates perspective (e.g. *'Lucy, I understand that you are very tired after work, but I feel it is unfair that I have to do all the cleaning by myself, and I think you should help with the cleaning'*), compared to simple requests where no perspective is communicated (e.g. *'Lucy, I think you should help with the cleaning'*). The primary mechanisms provided in the literature to explain the underlying benefits of communicating perspective is that it fosters a greater sense of 'feeling heard', and mutual understanding (*Ames, 2008*; *Ames & Wazlawek, 2014*; *Bruneau & Saxe, 2012*; *Howieson & Priddis, 2011*; *Lee et al., 2015*). It also fosters a sense of openness, transparency, and honesty that maximises perceived politeness and minimises perceived hostility (*Howieson & Priddis, 2015*; *Ingerson, DeTienne & Liljenquist, 2015*; *Kellas, Willer & Trees, 2013*; *Kidder, 2017*; *Seehausen et al., 2012*). The primary mechanisms to explain a benefit of I-language over you-language are that I-language indicates a recognition of providing a specific point of view that is open for discussion (*Burr, 1990*), and that you-language can at times be perceived as accusatory (*Kubany et al., 1992a*, *1992b*). Additionally, you-language might foster an increased tendency for inward focus in one's partner (*Brunye et al., 2009*, *2011*; *Ditman et al., 2010*).

How other specific language (e.g. we-language, should-statements, and speaking in absolutes) interact with one another to influence perceptions during conflict is an important avenue for further study. However, for now, this research gives us a significant insight into how two of the most commonly cited language strategies interact with each

other (i.e. I-language and communicating perspective-taking). This research supports the recommendations made by scholars within communication (*Hargie, 2011*), developmental (*Bloomquist, 2012*; *Heydenberk & Heydenberk, 2007*), business (*Whitcomb & Whitcomb, 2013*), and legal (*Howieson & Priddis, 2015*; *Kidder, 2017*; *Moore, 2014*) fields of scholarship to make use of communicative strategies such as I-language *and* communicating perspective during conflict to minimise hostility.

### Funding
The authors received no funding for this work.

### Competing Interests
The authors declare that they have no competing interests.

### Author Contributions
- Shane L. Rogers conceived and designed the experiments, performed the experiments, analysed the data, contributed reagents/materials/analysis tools, prepared figures and/or tables, authored or reviewed drafts of the paper, approved the final draft.
- Jill Howieson conceived and designed the experiments, authored or reviewed drafts of the paper, approved the final draft.
- Casey Neame conceived and designed the experiments, performed the experiments, authored or reviewed drafts of the paper, approved the final draft.

### Human Ethics
The following information was supplied relating to ethical approvals (i.e. approving body and any reference numbers):

Edith Cowan University granted ethical approval to carry out the study (Ethical Application Ref: 15257).

### Data Availability
The raw data are provided in the Supplemental Files.

### Supplemental Information
Supplemental information for this article can be found online at http://dx.doi.org/10.7717/peerj.4831#supplemental-information.

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
