# Peer review of "I understand you feel that way, but I feel this way: the benefits of I-language and communicating perspective during conflict"

_PeerJ, doi:10.7717/peerj.4831_

## Round 0.1 · original submission · Major Revisions

I have now received two reviews on your manuscript and have read it myself. As you will see, both reviewers are positively inclined toward your work, and I share their view. I thank the reviewers for their work. While according to Reviewer 2 your manuscript is almost ready for publications, Reviewer 1 proposes many observations you should take into account in a revision.

I see two categories of items that you will need to address for a successful revision.

1) You will need to work on the introduction and discussion sections, better grounding them in the framework of current literature and specifying in a more compact and consistent way the concepts and variables you introduce. As to literature grounding, Reviewer 1 offers many excellent suggestions; I add that there is a wide literature on I/you perspective taking in embodied cognition, and I think the manuscript would benefit if you briefly revise it (e.g. articles by Brunyé et al., by Gianelli et al., Beveridge & Pickering, 2013, etc.).

2) you will need to clarify how the sample size was determined (see comments of Reviewer 1)

Thank your for submitting your interesting work to PeerJ

·

Basic reporting

The paper aims to analyse the impact that the use of the I/You language and the self/other perspective may have on the perception of the degree of aggressiveness of the sentences they are part of. The experiment design is based on the subjects rating the likelihood of a “defensive response” to statements containing different combination of the two independent variables. The eight possible statements resulting from the combinations of I/You language and self/other perspective are introduced in relation to hypothetical scenarios in which the relationship and the specific conflict between the two parties are described. The sentences submitted for rating are presented as hypothetical opening statements of an hypothetical subsequent discussion.


The paper is well organised and clearly written. I think it may be an interesting contribution to the study of communication strategies that may help in preventing and managing conflict in interpersonal communication.
In what follows, I offer several suggestions as to how the manuscript might be improved, and I hope the authors can make use of them in some way.


1 Although proposing a well organised review of the literature in the three introductory sections, I think that the definition of the two main concepts (indipendent variables) - I/you language and self/other perspective - should be clarified and better introduced in relation to the previous literature. In particular, the definition of “communicating perspective” at lines 61- 64 may be misleading; the author defines it as including the communication of one’s own perspective and the understanding of the other’s perspective, but in Kellas et al. work, that the authors cite, “communicated perspective taking correspond to behaviours that communicate that one has put him/herself in the other’s shoes” (p. 327). I suggest the authors to discuss this definition and to better explain their choice to operationalize the communication of perspective-taking through the use of “self-oriented versus other-oriented communicated perspective” (see lines 129-130).

2 Between lines 75 and 87, the authors introduce other communicative and relational dimensions, such as “perceived understanding” , “feeling understood”, the matter of the communicative modality (oversimplified in verbal VS non-verbal) via which to communicate perspective, which all seem not to be relevant for the development of the paper. I suggest to eliminate all these brief hints to scarcely relevant dimensions, although addressed in previous studies.

3 Between lines 89 and 92, the authors mention assertiveness as linked to the self-oriented communicated perspective (see Kubany et al. on this point). Although in the literature assertiveness is associated to mitigation of aggressiveness (as Kubany et al. state at p. 337, assertiveness is the expression of opinions… in ways that respect the others…), at lines 90-92, the authors mainly associate it to anxiety and uncertainty, which are two more variables they introduce and abandon immediately. I think that at this point it would be more appropriate to elaborate on the relation between assertiveness and aggression.

5 At lines 101-103: the introduction of the I/you language variable and the examples the authors propose remind Kubany et al. distinction between “assertive I – messages” and “accusatory You-statements” (p. 339). Indeed, looking at the statements in table 1, some of them are clearly accusatory (i.e.“It is unfair that I have to do all the cleaning”), but the action –level (what type of action does a statement accomplish) is not taken into consideration. I think the authors should explain this lack, also in relation to Kubany et al. work.

6 I think that the authors risk to conceive and to use the I/you language category in a rather oversimplified way. Indeed, by isolating it from the larger discursive context within which it occurs, they seem to oversimplify its function and to predict its effect in an overly deterministic way.
I think that the communicative function that the I/you language component may have should be evaluated in relation to the type of discursive actions within which it occurs. In the statements that constitute the experimental material, the I/you language component and the self/other perspective component as well, are part and contribute to the realization of different actions such as giving accounts, allocating blame, criticizing or requesting to do. For this reason, I think that the relationship between the two components and the type of action they are part of should be better controlled in the experimental material and be taken into account in the analysis.

Experimental design

I illustrate below some aspects of the experimental design that I find problematic.

1 As I already said in the previous comments, one of the weaknesses of the experimental design is not taking into consideration the type of action accomplished by the sentences. This is true for the preparation of the materials and for the analysis as well, and I think this is even more problematic as the subjects are asked to rate which action (more or less defensive) is most likely to be produced in response to the sentences they are evaluating. In the experimental design, the response is evoked in terms of a more or less defensive responsive action, where “defensive” refers precisely to the action level.

2 The definition of what is a defensive response in terms of “reacting negatively or in a standoffish manner” (see the instruction at the end of each scenario) seems to me too vague.
I think that two alternatives would have been possible: to ask the subjects to directly rate de degree of aggressiveness and/or hostility of the sentences or to ask them to choose a response in a list of possible responses with different defensiveness degrees. I would like the authors discuss their experimental choice in the light of possible alternatives.

3 I am wondering if the presentation of the eight alternatives may affect the subjects’ rating in comparative terms. Could the authors discuss this point?

4 In the instruction, the reference to the defensive character of the hypothetical response may affect the subjects’ interpretation of the sentences by inducing an interpretative bias which could be paraphrased as “searching for aggressiveness/hostility”. I would like the authors discuss this point.

5 As also the authors state in the “Limitations” section, the experimental design is scarcely interactive. One of the alternatives proposed in 2 (asking the subjects to rate possible responsive actions) could be a partial, although I think still weak, solution to this problem. Previous studies addressing similar research questions, that the authors mention (see Bippus & Young and Kubany et al.), use much more interactive experimental designs. Could the authors better explain their experimental choice, discussing its advantages and disadvantages?

6 I find problematic the definition of the last sentence in the list as “No perspective….”.
Indeed “Lucy, I think you should help with the cleaning….” clearly conveys the speaker’s perspective. I would say that this is a request without the I/you language elements that, in the other sentences, contextualize the request, but I don’t think this is a case of “no communication of perspective”. I would ask the authors to clarify this point.

Validity of the findings

Generally speaking, the discussion and conclusion could be improved by making them more coherent with the previous theoretical and methodological sections of the paper. Indeed, in these last sections, while describing and discussing the results, the authors introduce some new analytical dimensions which were not introduced, nor discussed previously. This lack of consistency makes the description and discussion of the findings rather weak. I offer some suggestions in order to improve these last sections of the paper.

Section “Discussion”

1 Line 195, the “perceived impact”: I suggest to be more precise here. Indeed, the perceived impact is assessed only indirectly, through the rating of the likelihood of a defensive reaction.


2 Between lines 201 and 207, the authors introduce a new dimension: simple VS complex statements. As this variable was not introduced, nor used to describe the experimental material , I suggest the authors to be more consistent in the use of the analytical categories.
I think that the category “simple VS complex” could be related to the matter of degree of contextualization of the I/you language dimension, which I addressed in my comment 6 in “Basic reportings”.


3 At line 220 “the perspective of the self and other is the best strategy for opening a conflict discussion”. The statements listed in Table 1 are introduced as “statements that A could use to begin a conversation with B to resolve the issue” (see the instruction s that precede the list of statements). The statements to rate are described in two different manners: in the description of the scenario, the statement opens a conversation, while in the discussion of the results the authors refer to the statements as opening a conflict discussion. Whereas the conflict could or could not develop, according to the interlocutor’s interpretation of the previous statement, I find it problematic the use of the expression “conflict discussion”.

Section Conclusions

4 Line 264 “forms of language”: this is too generic. I suggest the authors to use a more appropriate definition. What follows is just a suggestion, which could be verified according to the literature: the I/you language and the communication of perspectives may be defined linguistic device (the first), communicative strategies (the second).

5 line 272: “communicative acts”. This seems very problematic. The authors never used, nor mentioned the speech act theory, nor they have addressed or analysed the sentences in terms of actions’ accomplishment (see my previous comments on this point). I suggest to clarify at what level and which linguistic-discursive dimension the I/You language and the perspective communication belong.

6 At lines 265-271 “Specifically the study demonstrates how I-language becomes more beneficial for minimising hostility when one also communicates perspective compared to simple requests where no perspective is communicated”. I suggest the authors to elaborate on this point and try to make some hypothesis to explain it.

Reviewer 2 ·

Basic reporting

The present paper presents a rather straightforward study. The authors present it in a concise but sufficiently detailed manner. The raw data are shared as raw_data_SPSS.sav, which is a property format of SPSS. In order to grant access from a larger audience, I would have recommended the use of another format (e.g. .csv)

Experimental design

The experimental design is well reported. The paper does not state whether the sample size of this study was determined a priori, or whether any stopping rules during data collection were used. Please report any information that might help the reader to assess how the sample size was determined.

Validity of the findings

Results are statistically sound and reported with sufficient clarity.
Conclusions are well connected to the original hypotheses and the reviewed literature. Limitations and future extensions are also taken into account.

---

## Round 0.2 · accepted · Accept

Thank you for carefully revising the paper. I am happy to inform you that your paper has now been accepted for publication on PeerJ.

#